# Velefibrinase: A Marine-Derived Fibrinolytic Enzyme with Multi-Target Antithrombotic Effects Across Diverse In Vivo Models

**DOI:** 10.3390/biomedicines13061277

**Published:** 2025-05-23

**Authors:** Yuting Zhou, Bo Yu, Chaoyin Xie, Manli Liu, Tiantian Long, Zhiqun Liang

**Affiliations:** 1Food Safety and Nutrition Experimental Teaching Demonstration Center, Department of Food Science and Engineering, Xinjiang Institute of Technology, Aksu 843000, China; 2023055@xjit.edu.cn (Y.Z.); 2024055@xjit.edu.cn (C.X.); 2022095@xjit.edu.cn (M.L.); 2022151@xjit.edu.cn (T.L.); 2Guangxi Microorganism and Enzyme Research Center of Engineering Technology, College of Life Science and Technology, Guangxi University, 100 Daxue Road, Nanning 530004, China; 1908401010@st.gxu.edu.cn

**Keywords:** fibrinolytic enzyme, thrombolytic, anticoagulation, neuroprotective

## Abstract

**Background/Objectives:** Thrombotic diseases (TDs), currently the number one killer worldwide, account for the highest mortality rate globally. In this study, we evaluated the antithrombotic efficacy of Velefibrinase, a marine bacteria-derived fibrinolytic enzyme, across multiple animal models. **Results:** The results demonstrated that Velefibrinase prolonged bleeding time (BT) and clotting time (CT), reduced mortality and thrombosis, relieved pulmonary alveolar structure degeneration in an acute pulmonary thromboembolism model, and inhibited carotid artery thrombosis and endothelial tissue damage in a rat model of FeCl_3_-induced carotid arterial thrombosis. Moreover, Velefibrinase reduced cerebral ischemia volume and ameliorated neurological deficits in a cerebral ischemia/reperfusion (I/R) injury model in rats. The putative underlying mechanisms were found to involve the inhibition of platelet aggregation and coagulation, along with the modulation of oxidative stress and inflammation levels. **Conclusions:** These results revealed that Velefibrinase exerts a notable thrombosis-preventive effect by interacting with multiple targets, thereby breaking the vicious cycle involving inflammation, oxidative stress, and thrombosis.

## 1. Introduction

Thrombotic diseases (TDs), including stroke, pulmonary embolism, and myocardial infarction, account for approximately 32% of global mortality, making them the leading causes of death worldwide [1,2]. An imbalance between coagulation and fibrinolysis is a major pathophysiological process in TDs, leading to the overproduction of thrombin and the uncontrolled hydrolysis of fibrinogen (FIB) to fibrin, its insoluble form [3,4]. Insoluble fibrin fibers subsequently combine with erythrocytes and activated platelets and precipitate in blood vessels, ultimately forming blood clots and thrombi, which present significant clinical challenges. However, thrombolytic agents can degrade fibrin fibers and lyse thrombi, thereby restoring blood flow in ischemic areas [5], rendering thrombolytic therapy the main treatment method for TDs in clinical practice.

Typical thrombolytic agents are generally divided into two categories. One category comprises plasminogen activators, such as tissue plasminogen activators (t-PAs) and urokinase, which convert plasminogen into plasmin, a hydrolase of numerous proteins, including fibrin. Despite their efficacy, these agents are associated with undesirable side effects, such as excessive bleeding, hemorrhage, and vomiting, and are also expensive [6]. The other category encompasses plasmin-like fibrinolytic enzymes, such as lumbrokinase and nattokinase, which can directly act on fibrin in thrombi, rapidly and completely dissolving them. In contrast to plasminogen activators, they pose a lower risk of bleeding and hemorrhage [7,8]. Notably, these fibrinolytic enzymes can be obtained from terrestrial sources, such as fermented food [9], traditional herbal sources [3], snake venom [10], earthworm [11], and mushroom [12]. While much of the attention to date has focused on fibrinolytic enzymes from these sources, marine environments remain largely untapped [13]. Considering the wide range of marine environments, fibrinolytic enzymes from marine sources may have unique biochemical properties and more advantages than those from terrestrial ones [14]. Meanwhile, although substantial research has been undertaken on fibrinolytic enzymes over recent decades, related studies have mainly focused on their purification and characterization, as well as the evaluation of their anti-thrombotic ability in vivo using a single animal thrombus model. However, only a few studies have systematically evaluated their anti-thrombotic potency in vivo. In our previous study [15], we purified and characterized a fibrinolytic enzyme, Velefibrinase, from *Bacillus velezensis* Z01, which was isolated from marine mud in the South China Sea. Velefibrinase demonstrated an efficient thrombolytic ability, suppressed platelet aggregation, and ameliorated blood coagulation in vitro. Moreover, Velefibrinase was found to be a non-cytotoxic, non-hemolytic enzyme with a superior anti-thrombotic ability in a κ-carrageenan-induced rat tail thrombosis model.

Building on our earlier findings, in the present study, we evaluated the in vivo anti-thrombotic efficacy of Velefibrinase using multiple animal models, including tail-tip transection, FeCl_3_-induced carotid artery thrombosis, collagen plus epinephrine-induced acute pulmonary thromboembolism, and middle cerebral artery occlusion and reperfusion (MCAO/R)-induced cerebral ischemia/reperfusion (I/R) injury. Furthermore, we conducted a preliminary investigation into the mechanisms by which Velefibrinase ameliorates cerebral I/R injury.

## 2. Materials and Methods

### 2.1. Materials

Velefibrinase (molecular weight: 32.3 kDa) was purified as previously described [15]. A protein quantification kit (Bradford assay) was purchased from TransGen Biotech (Beijing, China). Kits for the detection of malondialdehyde (MDA), superoxide dismutase (SOD), catalase (CAT), glutathione peroxidase (GSH-Px), tumor necrosis factor-alpha (TNF-α), interleukin-1 beta (IL-1β), thromboxane B2 (TXB_2_), and 6-keto-PGF_1a_ were obtained from Beyotime Biotech (Shanghai, China). Collagen, epinephrine, ketamine, xylazine, paraformaldehyde, adenosine diphosphate (ADP), and tetrazolium chloride (TTC) were purchased from Sigma-Aldrich (St. Louis, MO, USA). All other chemicals were of analytical or sequencing grade.

### 2.2. Animals

Specific pathogen-free male Sprague–Dawley (SD) rats (250–280 g) and male BALB/c mice (20–30 g) were acquired from the Guangxi Medical University Laboratory Animal Center. All animals were housed in a temperature- and humidity-controlled laboratory (temperature: 20 ± 1 °C, humidity: 60% ± 10%) under artificial light (12 h/12 h light/dark cycle) and had free access to food and water. All experimental protocols were performed according to the “Principles of Laboratory Animal Care” (NIH publication no. 8023, revised 1978) and were approved by the Animal Ethics Committee of Guangxi University (Permission no.: GXU-2021-046).

### 2.3. Assessment of Tail Vein Bleeding Time and Clotting Time

To determine the bleeding time (BT) and clotting time (CT), the tail-tip transection model was adopted, as described by Gedik [16] and Waghmare [17], with slight modifications. Briefly, the mice were divided into five groups, namely, a control group (isotonic saline), a positive control group (0.44 mg/kg urokinase), and three Velefibrinase treatment groups (low dose: 0.22 mg/kg, middle dose: 0.44 mg/kg, and high dose: 0.88 mg/kg). Each group of mice was administered with the corresponding dose for 3 consecutive days by tail vein injection. One hour after the last injection, the tail was cut 3–4 mm from the tip with sharp scissors. Then, the tail wound was gently touched with filter paper at regular intervals, and the time it took for no blood to appear on the paper was recorded as the BT. The CT was measured using Wright’s capillary glass tube method. One hour after the last administration, blood was collected from the tail wound using a clean capillary glass tube. Subsequently, the tube was gradually fractured at regular intervals, and the CT was recorded when a blood streak appeared at the fracture site.

### 2.4. Evaluation of the In Vivo Antithrombotic Activity of Velefibrinase in a Model of Acute Pulmonary Thromboembolism

The collagen plus epinephrine-induced pulmonary thromboembolism model was generated as detailed in Shi et al. [18]. Initially, the mice were randomly divided into six groups based on body weight—a sham group (isotonic saline, i.v.), a model group (isotonic saline, i.v.), a positive control group (0.44 mg/kg urokinase, i.v.), and three Velefibrinase treatment groups (low dosage: 0.22 mg/kg, middle dosage: 0.44 mg/kg, and high dosage: 0.88 mg/kg; i.v.). Each group of mice was administered with the corresponding dose for 5 consecutive days. One hour after the last administration, except for the sham group, the mice were challenged with 0.5 mL of a mixture containing collagen (1.5 mg/kg) and epinephrine (0.5 mg/kg) via an intravenous tail vein injection. Mouse mortality was monitored for 15 min, and survival rates were recorded as percentages for each treatment group. At the end of the experimental period, the surviving animals were euthanized with an overdose of anesthesia.

Complete lungs were removed from each mouse and cleaned with isotonic saline. Then, all lung lobes were wiped with filter paper, their wet weight was measured, and the lung coefficient was calculated as the wet weight-to-body weight ratio [19]. Subsequently, the lungs were carefully dissected, fixed in 4% paraformaldehyde overnight, embedded in paraffin, and cut into 5 μm thick sections using a microtome (RM2235, Leica, Weztlar, Germany). Paraffin-embedded sections were stained with hematoxylin and eosin (H&E) and observed under an optical microscope (COVER-015, OLYMPUS, Tokyo, Japan) for the evaluation of the antithrombotic effect of Velefibrinase.

### 2.5. In Vivo Evaluation of the Antithrombotic Activity of Velefibrinase in a Carotid Artery Thrombosis Model

A rat model of FeCl_3_-induced carotid artery thrombosis was established as previously described [20], with some modifications. Briefly, rats were randomly divided into six groups—a sham group (isotonic saline, i.v.), a model group (isotonic saline, i.v.), a positive control group (0.34 mg/kg urokinase, i.v.), and three Velefibrinase treatment groups (low dosage: 0.17 mg/kg, middle dosage: 0.34 mg/kg, and high dosage: 0.68 mg/kg). Each group of rats was administered with the corresponding dose for 5 consecutive days. One hour after the last administration, all the animals were anesthetized with ketamine (80 mg/kg, i.p.) and xylazine (10 mg/kg, i.p.), and a segment of the left carotid artery was exposed by blunt dissection. Thrombosis was induced by applying filter paper (8 × 8 mm) saturated with 5% FeCl_3_ solution to the adventitial surface of the carotid artery for 3 min. In the sham group, the FeCl_3_ solution was replaced with isotonic saline, while all other procedures were the same. Subsequently, the area of the carotid artery treated with FeCl_3_ was carefully dissected in each rat. Excess water was removed with absorbent paper, and weight was measured. The tissue was then fixed in 4% paraformaldehyde overnight, paraffin-embedded, sectioned, stained with H&E, and finally observed under an optical microscope (COVER-015, OLYMPUS, Tokyo, Japan) for the evaluation of the antithrombotic effect of Velefibrinase. Quantitative thrombus evaluation was performed using ImageJ 1.44p software (National Institutes of Health, Bethesda, MD, USA).

### 2.6. Cerebral I/R Injury Model

Cerebral I/R injury was induced in rats by MCAO/R, as previously detailed [21], with slight modifications. Briefly, isotonic saline, urokinase, and Velefibrinase (0.17, 0.34, or 0.68 mg/kg) were administered via a tail vein injection for 5 consecutive days. One hour after the last administration, all the animals were anesthetized with ketamine (80 mg/kg, i.p.) and xylazine (10 mg/kg, i.p.). A silicone-coated 4/0 monofilament nylon suture with a rounded tip (Cinontech Co. Ltd., Beijing, China) was gently introduced into the internal carotid artery (ICA) through the external carotid artery (ECA) and eventually advanced to occlude the MCA. After 2 h of occlusion, blood flow was restored (reperfusion) by carefully withdrawing the monofilament. In the sham group, the same surgical procedures were performed, except for MCA occlusion

### 2.7. Evaluation of Neurological Deficits and Cerebral Infarct Volume in the Cerebral I/R Injury Model

After 24 h of reperfusion, neurological deficits were scored using a previously described method [22], with slight modifications. The scoring protocol was as follows: 0, normal walking; 1, unable to walk straight and mild forelimb weakness; 2, circling toward the paretic side and severe forelimb weakness; 3, falling to the paretic side; 4, no spontaneous walking and a depressed level of consciousness; and 5, death.

The cerebral infarct volume was determined by TTC staining. All rats were euthanized using an overdose of anesthesia, followed by cervical dislocation and decapitation. The brains were rapidly removed, frozen at −20 °C for 30 min, sliced into 3 mm thick sections, immersed in 2% TTC solution at 37 °C for 30 min, and fixed in 4% paraformaldehyde solution overnight. The brain sections were imaged using a digital camera (WL-D88, Canon, Tokyo, Japan). The infarct area on each slice was quantified using ImageJ 1.44p software (National Institutes of Health).

### 2.8. Determination of the Antiplatelet and Anticoagulation Activity of Velefibrinase in a Cerebral I/R Injury Model

For each group, blood samples were drawn from the left abdominal aorta, and the effect of Velefibrinase on ADP-induced platelet aggregation in cerebral I/R injury was evaluated using the method described by Cheng et al. [23]. Briefly, the platelet count was determined using a hemocytometer. Then, platelet-rich plasma (PRP) was diluted with platelet-poor plasma (PPP) to a concentration of 3 × 10^8^ cells/mL. The PRP (300 μL/tube) was preheated for 10 min at 37 °C, mixed with 5 μmol/L of ADP, and then assessed for platelet aggregation using an aggregometer (SC-40, STEELLEX & Co., Beijing, China) at 37 °C with constant stirring.

The effect of Velefibrinase on the coagulation cascade was evaluated by detecting the coagulation parameters. Activated partial thromboplastin time (aPTT) reflects the intrinsic coagulation pathway, mainly influenced by factors VIII, IX, XI, and XII. Prothrombin time (PT) is associated with the extrinsic coagulation pathway, primarily regulated by coagulation factors V, VII, and X. Meanwhile, thrombin time (TT) measures the ability of fibrinogen to convert into fibrin, indicating the functional status of the terminal stage of coagulation. The method used to determine the in vivo anticoagulation potential of Velefibrinase followed that described by Kim et al. [24]. PPP was prepared by the centrifugation of blood samples from each group at 3000 rpm for 10 min at 4 °C. The PPP (300 μL per tube) was preheated for 10 min at 37 °C. The effect of Velefibrinase on aPTT, PT, TT, and the FIB level in the plasma of animals with cerebral I/R injury was determined using a coagulometer (LG-PABER-I, STEELLEX & Co.) according to the manufacturer’s instructions.

### 2.9. Detection of the Levels of TXB_2_/6-keto-PGF_1a_, Oxidative-Stress-Related Factors, and Inflammatory Cytokines in Plasma

Blood was collected and plasma was prepared using the above-mentioned method. The levels of TXB_2_, 6-keto-PGF_1a_, oxidative stress markers (SOD, MDA, CAT, and GSH-Px), and inflammatory cytokines (TNF-α and IL-1β) in plasma were determined using biochemical and ELISA assay kits, respectively, following the manufacturers’ instructions. The OD value was measured with a microplate reader.

### 2.10. Statistical Analysis

All experiments were conducted in triplicate and all data are expressed as means ± standard deviation (SD). The results were analyzed using one-way analysis of variance (ANOVA) or Tukey’s test. *p*-values of less than 0.05 were considered to be statistically significant.

## 3. Results

### 3.1. Velefibrinase Prolonged Blood BT and CT

As shown in Table 1, the BT and CT of thd mice in the control group were 173.17 s and 122.67 s, respectively. When mice were injected with urokinase at a dosage of 0.44 mg/kg, the BT and CT significantly increased to 134.83 s (*p* < 0.05) and 195.67 s (*p* < 0.01), respectively, compared to the control group. In comparison with the control treatment, low-dose (0.22 mg/kg) Velefibrinase significantly prolonged the CT by 10.32% (*p* < 0.05), but did not significantly influence the BT. In the middle-dose Velefibrinase treatment group, both the BT and CT were significantly prolonged relative to those recorded in the control group; however, no significant differences were observed when compared with the low-dosage Velefibrinase group or the urokinase group at the same dosage. In the high-dosage Velefibrinase group, the duration of BT (212.33 s) and CT (166.50 s) was further significantly prolonged, reaching the highest values observed.

### 3.2. The Effect of Velefibrinase on Acute Pulmonary Thromboembolism

As shown in Table 2, compared with acute pulmonary thromboembolism model mice, mortality and the lung coefficient were significantly decreased in animals treated with Velefibrinase. The protection rate of middle-dose Velefibrinase was 50%, while that of urokinase was only 33.33%, which was equivalent to that of high-dosage Velefibrinase (0.88 mg/kg). In the control group, the lung coefficient of mice was 0.76. However, when mice were injected with collagen and epinephrine (model group), the average lung coefficient increased to 1.10. In contrast to the model group, the lung coefficient of mice in the low-dosage Velefibrinase group was significantly reduced by 14.13% (*p* < 0.01). Moreover, the lung coefficient continued to decline with increasing Velefibrinase concentrations, and at the Velefibrinase concentration of 0.88 mg/kg (high dosage), the lung coefficient dropped to 0.88. These findings indicate that Velefibrinase ameliorated acute pulmonary thromboembolism infarction in a dose-dependent manner. Additionally, while urokinase treatment significantly reduced the lung coefficient (0.95, *p* < 0.01), its therapeutic efficacy did not differ significantly from that of low-dose Velefibrinase.

The results of the pathological evaluation of lungs with acute pulmonary thromboembolism are presented in Figure 1. Under the microscope, mice in the sham group displayed a normal, thrombus-free structure. In contrast, mice from the model group showed marked pulmonary thromboembolism, accompanied by alveolar damage and alveolar wall thickening. Compared with model mice, animals treated with low-dose Velefibrinase manifested a reduction in thrombosis, while alveolar damage and thickening were alleviated. Notably, the efficacy of low-dose Velefibrinase was comparable to that of urokinase. In the middle-dosage and high-dosage Velefibrinase treatment groups, the number of thromboembolisms was further decreased. These results indicate that Velefibrinase can prevent thrombus formation in acute pulmonary thromboembolism model mice.

### 3.3. Velefibrinase Relieved the Infarction Resulting from FeCl_3_-Induced Carotid Arterial Thrombosis

To investigate the effect of Velefibrinase on arterial thrombosis, we employed a FeCl_3_-induced carotid arterial thrombosis model in rats. As shown in Figure 2c, the average wet thrombus weight in the model group was 2.82 mg. Notably, the average wet thrombus weight was significantly lower (*p* < 0.05) in all the Velefibrinase treatment groups than in the model group, and the therapeutic effect of Velefibrinase was dose-dependent. In addition, no difference in wet thrombus weight was observed between middle-dose Velefibrinase (1.51 mg) and urokinase (1.72 mg).

The pathology of right carotid artery thrombosis in each group is shown in Figure 2a. The rates of carotid artery thrombosis occlusion in the different groups were calculated using Image-Pro Plus 6.0 (Figure 2b). In the sham group, no thrombi were detected in the carotid artery. After treatment with 5% FeCl_3_ (model group), a tightly textured thrombus appeared in the carotid artery, occluding 80.68% of the blood vessel space. Moreover, severe damage, including the peeling of vascular endothelial tissue, was also observed in the model group. However, the carotid artery thrombosis occlusion rates in the Velefibrinase treatment groups were significantly decreased (*p* < 0.01) relative to that of the model group, and this effect was dose-dependent. Meanwhile, with increasing Velefibrinase concentrations, the thrombus texture became increasingly loose, accompanied by an increase in the fraction of voids and a marked alleviation of the vascular endothelial lesions seen in the model group. Urokinase (0.44 mg/kg) also significantly reduced the thrombus occlusion rate in the carotid artery to 42.32% (*p* < 0.01). Nevertheless, its therapeutic efficacy did not differ significantly from that of Velefibrinase at the same dose.

### 3.4. Velefibrinase Improved Brain Infarcts and Attenuated Neurological Deficits in MCAO/R Rats

To evaluate the neuroprotective effects of Velefibrinase on focal brain ischemic injury, rats were subjected to 2 h of MCAO followed by 24 h of reperfusion. Extensive infarction was detected by TTC staining in the cerebral cortical and subcortical areas over a series of sections of the ipsilateral hemisphere in rats subjected to MCAO/R (Figure 3a,b). A large ischemic area appeared on the right hemisphere in the model group, with the infarct area accounting for 21.72% of the whole brain. However, when the rats were pretreated with Velefibrinase, the cerebral infarct volumes showed a significant narrowing trend compared with those of the model group. Moreover, the cerebral infarct volume was markedly decreased to 4.53% of the whole brain. Urokinase also significantly decreased the infarct volume to 10.39% (*p* < 0.05), and while its therapeutic efficacy was better than that of the low-dose Velefibrinase (infarct volume: 11.54%), the difference was not significant.

In addition, neurological deficit scores were determined in cerebral I/R model rats (Figure 3c). The neurological deficits induced by MCAO/R (model group) were significantly attenuated in both the urokinase and Velefibrinase treatment groups. Furthermore, the ameliorative effect of Velefibrinase was found to be dose-dependent.

### 3.5. The Effect of Velefibrinase on Platelet Aggregation and the Levels of TXB_2_ and 6-keto-PGF_1a_ in MCAO/R Rats

The effects of Velefibrinase on platelet aggregation and the contents of TXB_2_ and 6-keto-PGF_1a_ are shown in Table 3. The platelet aggregation rate in the I/R group was 45.92%, which was 0.63-fold lower than that of the sham group (74.97%). However, Velefibrinase treatment effectively inhibited platelet aggregation in vivo and in a dose-dependent manner. Compared with the I/R group (45.92%), both middle-dose (61.68%) and high-dose (56.70%) Velefibrinase significantly reduced the platelet aggregation rate.

The plasma levels of TXA_2_ and PGI_2_ in MCAO/R-induced cerebral I/R injury model rats were estimated by determining the concentrations of the respective stable hydrolysis products (TXB_2_ and 6-keto-PGF_1a_). Compared with the I/R group, the plasma concentrations of TXB_2_ were effectively downregulated (by 12.31%; *p* < 0.05) while those of 6-keto-PGF_1a_ were upregulated (by 37.60%; *p* < 0.01) in the low-dosage Velefibrinase group. In addition, no significant difference in the level of 6-keto-PGF_1a_ was detected between the high-dosage Velefibrinase and urokinase groups. Moreover, Velefibrinase treatment improved the 6-keto-PGF_1a_-to-TXB_2_ ratio in a dose-dependent manner. Notably, the improvement in the 6-keto-PGF_1a_-to-TXB_2_ ratio achieved with urokinase was comparable to that observed with middle-dose Velefibrinase, with no significant difference between the two treatments (*p* > 0.05).

### 3.6. The Effect of Velefibrinase on Coagulation Parameters in MCAO/R Rats

The effects of Velefibrinase on coagulation parameters—aPTT, PT, TT, and FIB—were determined, and the results are displayed in Table 4. Compared with the I/R group, aPTT and TT were significantly prolonged (*p* < 0.05) in the low-dosage Velefibrinase group, accompanied by a notable reduction in FIB content (*p* < 0.05). Moreover, the PT was markedly increased to 2.69 s in the middle-dosage Velefibrinase group (*p* < 0.05). Although the values for aPTT, PT, and TT in the middle-dosage Velefibrinase group were higher than those recorded for urokinase, the differences were not significant. In the high-dosage Velefibrinase group, the aPTT, PT, and TT values were further increased to 37.77 s, 2.79 s, and 31.98 s, respectively, and the FIB concentration showed a dose-dependent decrease, reaching a value of 4.02 g/L, which did not differ significantly from that of the sham group.

### 3.7. Velefibrinase Ameliorated Oxidative Stress and Reduced Inflammatory Cytokine Levels in MCAO/R Rats

The effect of Velefibrinase on oxidative stress is shown in Table 5. Compared with the sham group, the I/R group exhibited significant reductions in SOD, CAT, and GSH-Px levels by 66.28%, 63.16%, and 66.26%, respectively (all *p* < 0.01), while MDA levels showed a marked 1.04-fold increase (*p* < 0.01). Compared with the I/R group, the CAT and GSH-Px concentrations were significantly increased by 30.28% and 29.09% (*p* > 0.05), respectively, in the low-dosage Velefibrinase group. Furthermore, middle-dose Velefibrinase treatment resulted in significant improvements in oxidative stress markers, with increased SOD levels and decreased MDA levels. Notably, except for CAT, the efficacy of urokinase in improving oxidative stress levels was comparable to that of Velefibrinase at the same dose.

We further evaluated the effects of Velefibrinase on TNF-α and IL-1β contents, and the results are shown in Table 6. Compared with the sham group, in rats of the I/R treatment group, the plasma levels of TNF-α and IL-1β were significantly increased up to 65.00 pg/mL and 293.56 pg/mL, respectively. In comparison, the concentrations of TNF-α and IL-1β in the low-dosage Velefibrinase group were significantly decreased by 11.18% and 13.84% (*p* < 0.05), respectively. Furthermore, as the dose of Velefibrinase increased to the middle dose, the levels of TNF-α and IL-1β were further and more significantly reduced, with its efficacy being comparable to that of urokinase.

## 4. Discussion

Although a large number of fibrinolytic enzymes derived from natural sources have been purified and characterized, few studies have systematically evaluated their anti-thrombotic potency in vivo. In this study, the antithrombotic ability of Velefibrinase, a marine fibrinolytic enzyme, was evaluated in different animal models, with the results showing that Velefibrinase can inhibit thrombosis. In addition, Velefibrinase was found to break the vicious cycle of inflammation, oxidative stress, and coagulation, thereby ameliorating the infarction caused by cerebral I/R injury.

BT and CT are commonly employed methods for assessing the risk of bleeding. In this study, when administered to mice at a low dose (0.22 mg/kg), Velefibrinase did not significantly prolong the BT. However, as the dosage of Velefibrinase gradually increased, both BT and CT exhibited a prolongation trend. Meanwhile, we also observed that urokinase could similarly prolong BT and CT, and its effect showed no significant difference compared to Velefibrinase at equivalent doses. This indicates that Velefibrinase is unlikely to cause severe bleeding, consistent with the findings by Choi et al. [20] and Gupta [25]. However, it should be particularly noted that the degree of prolongation of BT and CT induced by a high dose of Velefibrinase was significantly higher than that of urokinase. Therefore, Velefibrinase still retains a potential bleeding risk, and caution is required during its use. Future research should aim to define the optimal therapeutic window for balancing efficacy and safety, especially during the clinical translation phase. In addition, CT also serves as an indicator of coagulation factor effectiveness in the intrinsic pathway. Our previous research showed that Velefibrinase inhibits κ-carrageenan-induced thrombus formation in mice tails [15]. Currently, the prevailing view is that κ-carrageenan activates the Hageman factor (coagulation factor XII) in vivo, thereby initiating the intrinsic coagulation pathway and ultimately leading to thrombus formation [26]. Based on this, we speculate that Velefibrinase may prolong CT by inhibiting the Hageman factor or other coagulation factors in the intrinsic pathway. This observation is also consistent with our findings relating to coagulation parameters, specifically that Velefibrinase prolonged aPTT (Table 4), indicating its potential effect on intrinsic pathway factors (XII, VIII, IX, and XI). Velefibrinase can reduce FIB content and directly lyse fibrin clots, both of which contribute to the prolongation of CT. Therefore, whether Velefibrinase directly affects the activity of intrinsic coagulation factors requires further experimental verification.

Collagen and epinephrine are both potent platelet activators that can generate a significant number of microemboli within the pulmonary vasculature, leading to acute pulmonary thromboembolism and subsequent mortality in mice [27]. The resultant platelet aggregation and the production of numerous microemboli in the lungs are associated with significant increases in both lung wet weight and the lung coefficient, ultimately resulting in death. However, these parameters were significantly improved following Velefibrinase treatment. A previous in vitro study found that Velefibrinase inhibits the aggregation of washed platelets, as well as thrombosis [15], potentially explaining why it reduced the mortality rate and lung coefficient in mice in a dose-dependent manner. In the present study, we also found that Velefibrinase inhibited platelet aggregation in vivo (Table 3). In agreement with this, optical microscopy revealed that the number of thromboemboli was decreased in lung histological sections in a manner dependent on the Velefibrinase dose. Additionally, the pulmonary alveolar structure was significantly deteriorated in mice with acute pulmonary thromboembolism, accompanied by alveolar wall thickening; however, these effects were significantly attenuated following Velefibrinase treatment. The finding that Velefibrinase can improve the symptoms of acute pulmonary thromboembolism has also been observed with other fibrinolytic enzymes, which have already been applied in clinical practice [28].

Ferric-chloride-induced thrombosis in the carotid artery has been used extensively for the in vivo testing of new antithrombotic therapies. It is well known that ferric chloride diffuses through the vessel wall, resulting in the destruction of endothelial tissue and the generation of oxidative stress by free radicals, which, in turn, induces platelet aggregation and blood clot formation [29]. Here, we found that Velefibrinase not only protects endothelial tissue from ferric-chloride-induced injury, but also inhibits thrombus formation in the carotid artery in a dose-dependent manner. This might be attributed to its ability to prevent endothelial injury, avoiding the exposure of von Willebrand factor (vWF) and collagen in the subendothelial matrix, thereby inhibiting platelet aggregation and thrombus formation. Furthermore, the preservation of endothelial integrity can also reduce oxidative stress levels and suppress intravascular thrombosis [30]. Aligning with this result, Ge et al. [31] reported that a marine fibrinolytic enzyme inhibited ferric-chloride-induced carotid thrombosis by suppressing platelet aggregation and vascular constriction.

Thrombolytic therapy and mechanical thrombectomy are effective therapies for acute ischemic stroke. However, revascularization can lead to severe secondary damage due to I/R injury. This can ultimately worsen stroke outcomes and pose a serious threat to human life [32]. We found that after 24 h of I/R, Velefibrinase treatment reduced the cerebral infarction volume and alleviated neurological deficits in MCAO/R model rats. Concurrently, we also observed that Velefibrinase ameliorated the ischemic condition in the ischemic penumbra, but did not prevent injury in the ischemic core area. This implies that Velefibrinase can promote the repair of neuronal cells that are in a dormant or semi-dormant state within the ischemic penumbra, thereby effectively alleviating neurological deficits and exerting significant neuroprotective effects [33]. These results are consistent with an earlier clinical pilot study, in which it was found that nattokinase prevented stroke progression in patients with acute ischemic stroke and demonstrated a marked neuroprotective effect [34].

Neuroprotective measures against cerebral ischemic injury include the modulation of inflammation and oxidative stress, both of which are associated with the apoptosis or necrosis of neuronal cells [35]. I/R injury is followed by a massive accumulation of reactive oxygen species (ROS), which can trigger severe oxidative stress, promote complement activation on platelets, induce endothelial dysfunction, activate inflammatory transcription factors, promote the release of inflammatory signals, and lead to the production of inflammatory cytokines, including IL-6, IL-1β, and TNF-α. Additionally, excessive ROS can cause neuronal death and the release of nucleosides, resulting in the activation of purine receptors on microglia and macrophages. This leads to inflammatory cell aggregation, infiltration, and, consequently, a series of secondary tissue injuries [36]. A fibrinolytic enzyme derived from *Neanthes japonica* has been reported to possess neuroprotective properties, potentially due to its ability to inhibit lipid peroxidation and enhance the activity of endogenous antioxidant defense enzymes [37]. Additionally, Li et al. [38] demonstrated that the suppression of oxidative stress and inflammation can mitigate cerebral I/R injury. In this study, Velefibrinase reduced the plasma MDA levels in rats subjected to cerebral I/R, while simultaneously increasing the activities of endogenous antioxidant enzymes (SOD, CAT, and GSH-Px). Velefibrinase also reduced the plasma contents of TNF-α and IL-1β. This implies that the neuroprotective effect exerted by Velefibrinase following cerebral I/R injury may be achieved through the alleviation of oxidative and inflammatory stresses.

Additionally, patients with acute stroke often exhibit a hypercoagulable state and an elevated risk of thrombus formation, a state that is directly related to platelet activation and coagulation impairment [39]. It has been reported that nattokinase exerts a neuroprotective effect in the ischemic brain by inhibiting platelet aggregation and improving blood flow [40]. TXA_2_ and PGI_2_ are stable metabolites of TXB_2_ and 6-keto-PGF_1α_, respectively. A pathological imbalance between TXA_2_ and PGI_2_ can induce platelet aggregation and thrombosis, leading to myocardial infarction and cerebrovascular accidents [41]. The secretion of TXA_2_ can induce vasospasm and thrombosis, whereas PGI_2_ secretion can alleviate thrombosis. Our results showed that Velefibrinase is a potent inhibitor of platelet aggregation, as evidenced by platelet aggregation assays and the determination of the TXB_2_/6-keto-PGF_1α_ ratio in a cerebral I/R injury model. Moreover, our findings suggested that Velefibrinase can ameliorate blood coagulation disorders. According to the cascade hypothesis, aPTT is related to the intrinsic coagulation pathway, PT is associated with the extrinsic pathway, and TT and FIB are linked to the third stage of coagulation in plasma [42]. Our results showed that Velefibrinase significantly prolonged aPTT, PT, and TT and decreased the content of FIB, indicating that Velefibrinase has anticoagulant potential, breaking down any one of the blood coagulation factors involved in the intrinsic and/or extrinsic pathways of the coagulation cascade. This result is consistent with the study of Kurosawa et al. [39], who found that nattokinase reduces the activity of coagulation factors VII and VIII, thereby prolonging PT and aPTT, respectively.

## 5. Conclusions

In the present study, we demonstrated that Velefibrinase inhibits thrombosis and ameliorates cerebral I/R injury. Moreover, we found that Velefibrinase exerts its effects by inhibiting platelet aggregation and the coagulation system, along with modulating oxidative stress and inflammation levels. These findings indicate that Velefibrinase exerts thrombosis-preventive effects by interacting with multiple targets and breaking the vicious cycle involving inflammation, oxidative stress, and thrombosis.

## Figures and Tables

**Figure 1 biomedicines-13-01277-f001:**
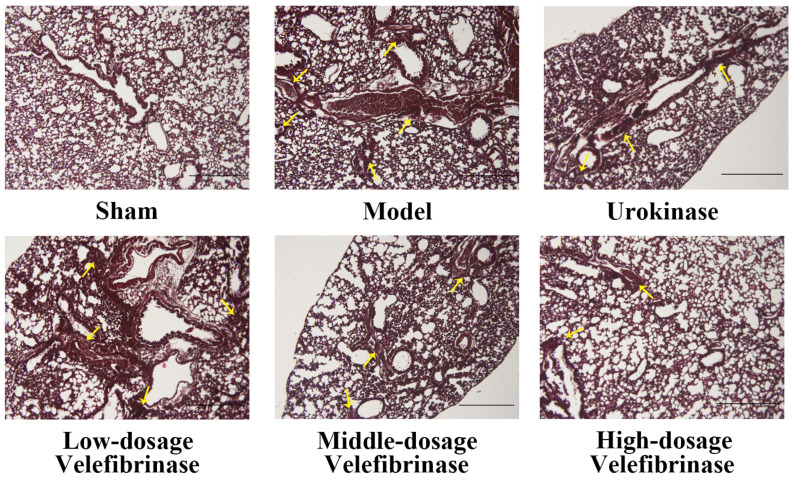
Photomicrograph (×100) of a hematoxylin and eosin (H&E)-stained mouse lung in acute pulmonary thromboembolism induced by collagen plus epinephrine (*n* = 6). Mice were intravenously injected with isotonic saline only (sham group), vehicle (1.5 mg/kg collagen and 0.5 mg/kg epinephrine, model group), vehicle + 0.44 mg/kg urokinase (urokinase group), vehicle + 0.22 mg/kg Velefibrinase (low-dosage group), vehicle + 0.44 mg/kg Velefibrinase (middle-dosage group), or vehicle + 0.88 mg/kg Velefibrinase (high-dosage group). The arrows indicate thrombus. Scale bar, 100 μm.

**Figure 2 biomedicines-13-01277-f002:**
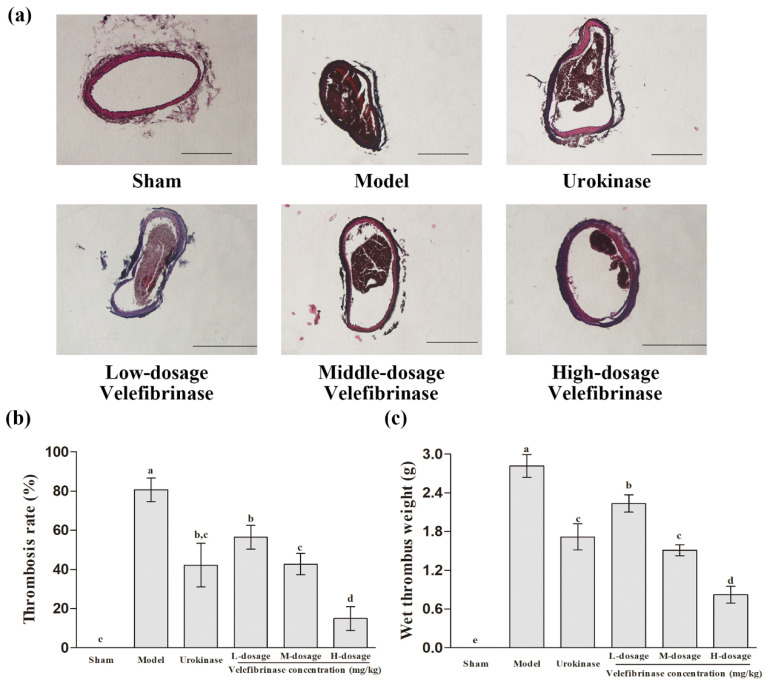
Velefibrinase protects against FeCl_3_-induced carotid arterial thrombosis. (**a**) A representative photomicrograph (×100) showing carotid arterial thrombosis. Sections were stained with hematoxylin and eosin (H&E). The right carotid artery was administered 5% FeCl_3_ (model group), 0.34 mg/kg urokinase (urokinase group), 0.17 mg/kg Velefibrinase (low-dosage group), 0.34 mg/kg Velefibrinase (middle-dosage group), or 0.68 mg/kg Velefibrinase (high-dosage group). Rats in the sham group underwent the same surgical procedures but without 5% FeCl_3_ treatment. Scale bar, 100 μm. (**b**) The occlusion rates of carotid artery thrombosis in the different groups. (**c**) The wet thrombus weight of the carotid artery in the different groups. Values are expressed as means ± SD (*n* = 6). Different letters indicate significant differences between groups (*p* < 0.05).

**Figure 3 biomedicines-13-01277-f003:**
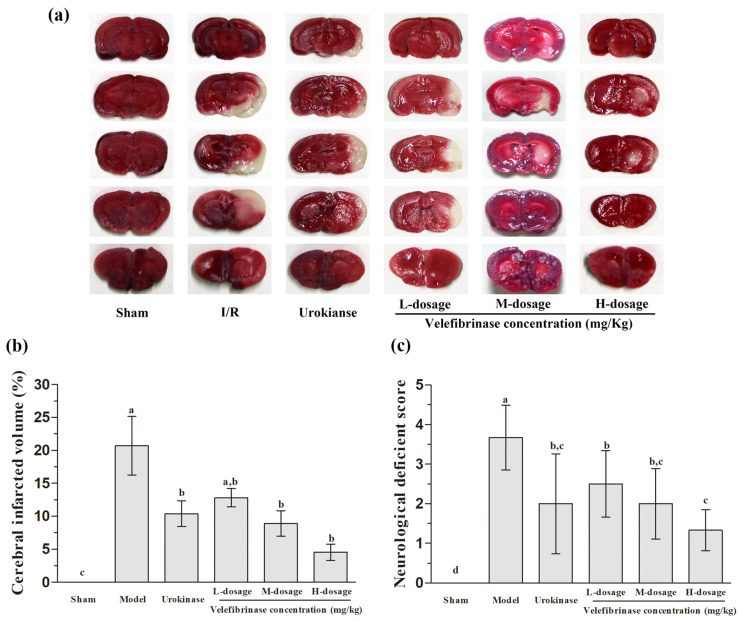
Velefibrinase reduced ischemia/reperfusion (I/R)-induced infarct volumes and neurologic deficits. (**a**) Images of TTC-stained cerebral slices 24 h after reperfusion. Following I/R treatment, rats were injected with isotonic saline (I/R group), 0.34 mg/kg urokinase (urokinase group), 0.17 mg/kg Velefibrinase (low-dosage group), 0.34 mg/kg Velefibrinase (middle-dosage group), or 0.68 mg/kg Velefibrinase (high-dosage group). Rats in the sham group underwent the same surgical procedures but without middle cerebral artery occlusion. (**b**) Velefibrinase attenuated I/R-induced neurological deficits 24 h after reperfusion. (**c**) TTC staining showing cerebral infarct volumes. Values are expressed as means ± SD (*n* = 6). Different letters represent significant differences between groups (*p* > 0.05).

**Table 1 biomedicines-13-01277-t001:** The effect of Velefibrinase on bleeding time (BT) and clotting time (CT) in mice.

Groups	Dose (mg/kg)	BT (s)	CT (s)
Control	-	173.17 ± 10.93 ^d^	122.67 ± 7.79 ^c^
Urokinase	0.44	195.67 ± 10.41 ^b,c^	134.83 ± 8.33 ^b^
L-dosage (Velefibrinase)	0.22	185.33 ± 12.94 ^c,d^	135.33 ± 9.42 ^b^
M-dosage (Velefibrinase)	0.44	200.50 ± 14.32 ^a,b^	141.50 ± 6.47 ^b^
H-dosage (Velefibrinase)	0.88	212.33 ± 13.50 ^a^	166.50 ± 7.82 ^a^

Each value is expressed as mean ± SD (*n* = 6). Different letters indicate significant differences between the groups by Tukey’s test (*p* < 0.05).

**Table 2 biomedicines-13-01277-t002:** The effect of Velefibrinase on mortality and the lung coefficient in acute pulmonary thromboembolism model mice.

Groups	Dose (mg/kg)	No. Dead/No. Tested	Protection (%)	Lung Coefficient
Control	-	0	100	0.76 ± 0.03 ^e^
Model	-	6	0	1.10 ± 0.06 ^a^
Urokinase	0.44	2	66.67	0.95 ± 0.04 ^b,c^
L-dosage (Velefibrinase)	0.22	3	50	0.99 ± 0.05 ^b^
M-dosage (Velefibrinase)	0.44	3	50	0.94 ± 0.03 ^c^
H-dosage (Velefibrinase)	0.88	2	66.67	0.88 ± 0.05 ^d^

Each value is expressed as mean ± SD (*n* = 6). Different letters indicate significant differences between the groups by Tukey’s test (*p* < 0.05).

**Table 3 biomedicines-13-01277-t003:** The effect of Velefibrinase on platelet aggregation and the contents of TXB_2_ and 6-keto-PGF_1a_ in a rat model of cerebral ischemia/reperfusion (I/R) injury.

Groups	Dose (mg/kg)	Platelet Aggregation Rate (%)	TXB_2_ (pg/mL)	6-keto-PGF_1a_ (pg/mL)	Ratio of TXB_2_ to 6-keto-PGF_1a_
Sham	-	45.92 ± 4.80 ^d^	279.36 ± 15.02 ^e^	542.27 ± 14.61 ^d^	1.94 ± 0.09 ^a^
I/R	-	74.97 ± 5.50 ^a^	595.36 ± 21.62 ^a^	448.73 ± 29.51 ^e^	0.75 ± 0.06 ^e^
Urokinase	0.34	64.48 ± 7.05 ^b^	466.45 ± 17.18 ^c^	669.36 ± 18.56 ^a,b^	1.44 ± 0.08 ^c^
L-dosage (Velefibrinase)	0.17	68.45 ± 6.06 ^a,b^	522.09 ± 18.28 ^b^	616.82 ± 21.49 ^c^	1.18 ± 0.05 ^d^
M-dosage (Velefibrinase)	0.34	61.68 ± 5.40 ^b,c^	467.91 ± 21.99 ^c^	652.09 ± 20.44 ^b^	1.40 ± 0.07 ^c^
H-dosage (Velefibrinase)	0.68	56.70 ± 8.91 ^c^	393.18 ± 13.88 ^d^	695.18 ± 17.74 ^a^	1.77 ± 0.04 ^b^

Each value is expressed as mean ± SD (*n* = 6). Different letters indicate significant differences between the groups (*p* < 0.05).

**Table 4 biomedicines-13-01277-t004:** The effect of Velefibrinase on four blood coagulation parameters in a rat model of cerebral ischemia/reperfusion (I/R) injury.

Groups	Dose (mg/kg)	aPTT (s)	PT (INR)	TT (s)	FIB (g/L)
Sham	-	40.82 ± 3.73 ^a^	2.92 ± 0.28 ^a^	34.13 ± 2.55 ^a^	3.85 ± 0.18 ^d^
I/R	-	23.22 ± 3.99 ^c^	2.27 ± 0.21 ^b^	23.27 ± 2.52 ^d^	5.33 ± 0.73 ^a^
Urokinase	0.34	27.83 ± 2.58 ^b^	2.67 ± 0.27 ^a^	27.38 ± 2.99 ^c^	4.82 ± 0.42 ^b^
L-dosage (Velefibrinase)	0.17	28.15 ± 2.34 ^b^	2.34 ± 0.21 ^b^	27.55 ± 3.61 ^c^	4.81 ± 0.32 ^b^
M-dosage (Velefibrinase)	0.34	31.30 ± 3.00 ^b^	2.69 ± 0.27 ^a^	29.37 ± 2.23 ^b,c^	4.29 ± 0.31 ^c^
H-dosage (Velefibrinase)	0.68	37.77 ± 4.47 ^a^	2.79 ± 0.26 ^a^	31.98 ± 2.97 ^a,b^	4.02 ± 0.19 ^c,d^

Each value is expressed as mean ± SD (*n* = 6). Different letters indicate significant differences between the groups (*p* < 0.05).

**Table 5 biomedicines-13-01277-t005:** The effect of Velefibrinase on plasma oxidative stress factors in a rat model of cerebral ischemia/reperfusion (I/R) injury.

Groups	Dose (mg/kg)	MDA (μmol/mg pro)	SOD (U/mg pro)	CAT (U/mg pro)	GSH-Px (U/mg pro)
Sham	-	2.39 ± 0.06 ^c^	38.09 ± 5.82 ^a^	15.42 ± 1.24 ^a^	6.52 ± 0.72 ^a^
I/R	-	4.87 ± 0.44 ^a^	12.94 ± 2.35 ^d^	5.68 ± 0.85 ^f^	2.20 ± 0.52 ^e^
Urokinase	0.34	4.21 ± 0.50 ^b^	18.21 ± 3.93 ^c^	8.76 ± 1.14 ^d^	3.58 ± 0.28 ^c^
L-dosage (Velefibrinase)	0.17	4.41 ± 0.54 ^a,b^	17.66 ± 3.73 ^c,d^	7.40 ± 0.98 ^e^	2.84 ± 0.31 ^d^
M-dosage (Velefibrinase)	0.34	4.18 ± 0.49 ^b^	22.03 ± 4.20 ^c^	10.07 ± 0.94 ^c^	3.45 ± 0.41 ^c^
H-dosage (Velefibrinase)	0.68	3.99 ± 0.30 ^b^	27.76 ± 3.98 ^b^	11.88 ± 0.84 ^b^	4.26 ± 0.39 ^b^

Each value is expressed as mean ± SD (*n* = 6). Different letters indicate significant differences between the groups (*p* < 0.05).

**Table 6 biomedicines-13-01277-t006:** The effect of Velefibrinase on plasma inflammatory factors in a rat model of cerebral ischemia/reperfusion (I/R) injury.

Groups	Dose (mg/kg)	TNF-α (pg/mL)	IL-1β (pg/mL)
Sham	-	47.83 ± 3.96 ^d^	90.74 ± 17.81 ^e^
I/R	-	65.00 ± 4.16 ^a^	293.56 ± 36.76 ^a^
Urokinase	0.34	57.19 ± 2.47 ^b^	221.66 ± 17.75 ^c^
L-dosage (Velefibrinase)	0.17	57.73 ± 4.89 ^b^	252.93 ± 23.79 ^b^
M-dosage (Velefibrinase)	0.34	52.50 ± 4.44 ^c^	190.99 ± 19.41 ^d^
H-dosage (Velefibrinase)	0.68	50.83 ± 2.60 ^c,d^	171.83 ± 18.81 ^d^

Each value is expressed as mean ± SD (*n* = 6). Different letters indicate significant differences between the groups (*p* < 0.05).

## Data Availability

The original contributions presented in this study are included in the article. Further inquiries can be directed to the corresponding author.

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
