# Peer review of "Velefibrinase: A Marine-Derived Fibrinolytic Enzyme with Multi-Target Antithrombotic Effects Across Diverse In Vivo Models"

_biomedicines, 2025, doi:10.3390/biomedicines13061277_

Round 1
Reviewer 1 Report
Comments and Suggestions for Authors
This is a study investigating the mechanism of action and utility of the bacteria-derived fibrinolytic enzyme, Velefibrinase, in combating thrombosis.
A few comments to consider:
- The abstract leads commenting that thrombotic disease has surpassed cancer in annual mortality. Thrombotic disease has always led cancer in terms of mortality, and so this misrepresents mortality statistics.
- The study starts by comparing the effect of Velefibrinase on bleeding time and clotting time as compared to urokinase, where the finding is that Velefibrinase increases both bleeding and clotting times above baseline and at higher doses, above urokinase; this in general suggests that the safety margin for Velefibrinase is less than urokinase, which will limit the translation of this as a drug candidate as this agent increases bleeding risk above that of urokinase.
- Next, the study investigates the effects of Velefibrinase in a model of acute pulmonary thromboembolism. There, Velefibrinase performs as well as urokinase in terms of efficacy.
- The effect of Velefibrinase was then compared in a model of FeCl3-induced CA thrombosis. The claim is that a dose-dependent effect was observed; were the statistics done comparing the groups with each other, or do the statistics only compare the data to the control? Justification for the claim of a dose-dependent effect should be clarified.
- The effect of Velefibrinase was then compared in a model of brain ischemic injury. Similar as above, justification for the claim of a dose-dependent effect should be clarified.
- The effects of Velefibrinase was observed on platelet aggregation was studied, where the agent performed similar to urokinase; Velefibrinase impaired platelet aggregation, thus suggesting that this agent is antithrombotic, yet unsafe. The same effect is observed in select coagulation parameters. Similar as above, justification for the claim of a dose-dependent effect should be clarified.
- The discussion centers around Hageman factor, which is FXII. The hypothesis that Velefibrinase inhibits FXII activation can be tested by measuring FXIIa-C1nh or FXIIa-AT complex formation. Or tested in vitro that this is possible in a purified system with a chromogenic substrate. The mechanism-of-action of how Velefibrinase would inhibit FXII activation or activity is unclear to the reviewer.
Author Response
Response to Reviewer 1 Comments
Point 1: The abstract leads commenting that thrombotic disease has surpassed cancer in annual mortality. Thrombotic disease has always led cancer in terms of mortality, and so this misrepresents mortality statistics.
Response 1: Thank you for your reminder. I have made modifications to this part, and the modifications are as follows: Thrombotic diseases (TDs) , currently the number one killer worldwide, account for the highest mortality rate globally.
Point 2: The study starts by comparing the effect of Velefibrinase on bleeding time and clotting time as compared to urokinase, where the finding is that Velefibrinase increases both bleeding and clotting times above baseline and at higher doses, above urokinase; this in general suggests that the safety margin for Velefibrinase is less than urokinase, which will limit the translation of this as a drug candidate as this agent increases bleeding risk above that of urokinase.
Next, the study investigates the effects of Velefibrinase in a model of acute pulmonary thromboembolism. There, Velefibrinase performs as well as urokinase in terms of efficacy.
Response 2: The high-dosage Velefibrinase group exhibited significantly prolonged bleeding time (BT) and clotting time (CT) compared to both the control group and the middle-dosage urokinase group. This result is similar to that of Wu et al., where the BT and CT of nattokinase and heparin also increased with the increase in dosage.
We fully acknowledge the reviewer’s concern about the safety margin of Velefibrinase. Although Velefibrinase at high doses shows a greater bleeding risk and a relatively smaller safety margin compared to urokinase, we do not believe this completely rules out its potential as a drug candidate. Future research could focus on developing novel therapeutic strategies, like sustained-release technology or combination therapy, to attempt to expand its safety margin while maintaining its antithrombotic efficacy.
Additionally, I have refined the discussion in this section as follows: However, it should be particularly noted that the degree of prolongation of the BT and CT by a high dose of Velefibrinase is significantly higher than that of urokinase. Therefore, Velefibrinase still retains a potential bleeding risk, and caution is required during its use. Future research should aim to define the optimal therapeutic window for balancing efficacy and safety, especially during the clinical translation phase.
In the case of acute pulmonary thromboembolism, the therapeutic effect of Velefibrinase was similar to that of urokinase. Currently, We are now optimizing its production via low-cost fermentation to make it as an affordable antithrombotic drug candidate.
Reference:
Wu H, Wang H, Li W, et al. Nattokinase-heparin exhibits beneficial efficacy and safety-an optimal strategy for CKD patients on hemodialysis [J]. Glycoconjugate journal, 2019, 36(2):93-101. DOI: 10.1007/s10719-019-09860-8.
Point 3: The effect of Velefibrinase was then compared in a model of FeCl3-induced CA thrombosis. The claim is that a dose-dependent effect was observed; were the statistics done comparing the groups with each other, or do the statistics only compare the data to the control? Justification for the claim of a dose-dependent effect should be clarified.
The effect of Velefibrinase was then compared in a model of brain ischemic injury. Similar as above, justification for the claim of a dose-dependent effect should be clarified.
The effects of Velefibrinase was observed on platelet aggregation was studied, where the agent performed similar to urokinase; Velefibrinase impaired platelet aggregation, thus suggesting that this agent is antithrombotic, yet unsafe. The same effect is observed in select coagulation parameters. Similar as above, justification for the claim of a dose-dependent effect should be clarified.
Response 3: The data were the statistics done comparing the groups with each other, not just against the control. Meanwhile, it is written below both the tables and the pictures that: Different letters indicate significant differences between the groups (p < 0.05).
According to the experimental results, the antithrombotic effect of Velefibrinase showed significant improvement with increasing dosage. Therefore, we conclude that it has a dose-dependent effect, which is consistent with the descriptions in other literatures.
In our previous report, we demonstrated that incubation of Velefibrinase with washed platelets showed no significant difference in lactate dehydrogenase leakage compared to the saline control group, indicating minimal platelet toxicity.
As for the therapeutic effect of Velefibrinase was similar to that of urokinase, we are currently developing a low-cost fermentation process to establish it as an affordable antithrombotic therapeutic option.
Reference:
Zhang Y, Li XP, Yang Q, et al. Antioxidation, anti-hyperlipidaemia and hepatoprotection of polysaccharides from Auricularia auricular residue [J].Chemico-Biological Interactions, 2021, 333. DOI:10.1016/j.cbi.2020.109323.
Zhou YT, Chen HZ, Yu B, et al. Purification and Characterization of a Fibrinolytic Enzyme from Marine Bacillus velezensis Z01 and Assessment of Its Therapeutic Efficacy In Vivo [J]. Microorganisms 2022, 10, 843. DOI: 10.3390/microorganisms10050843.
Point 4: The discussion centers around Hageman factor, which is FXII. The hypothesis that Velefibrinase inhibits FXII activation can be tested by measuring FXIIa-C1nh or FXIIa-AT complex formation. Or tested in vitro that this is possible in a purified system with a chromogenic substrate. The mechanism-of-action of how Velefibrinase would inhibit FXII activation or activity is unclear to the reviewer.
Response 4: We thank the reviewer for suggestion. While our current study has not yet included these specific mechanistic investigations, we will investigate these mechanisms in future research.
Reviewer 2 Report
Comments and Suggestions for Authors
In this paper, Zhou et al. investigate the fibrinolytic and antithrombotic activites of Velefibrinase, an enzyme derived from Bacillus velezensis. The study is relevant to the development of possible therapy against thrombotic diseases, which contribute to a large number of deaths globally. The Authors report on an overall evaluation of the impact of Velefibrinase in a variety of in vivo models that depicts its potential to prolong the bleeding and clotting times, prevent thrombosis, and relieve ischemia-induced injury. They further explore underlying mechanisms, suggesting that Velefibrinase acts by inhibiting platelet aggregation, modulating oxidative stress, and reducing inflammation. Notably, the authors discuss its effects multiple target action, stopping the thrombosis-inflammatory pathway. Though the molecular interactions are not yet entirely clear, this work offers a sound basis for upcoming investigations of marine-derived enzymes as sophisticated antithrombotic drugs.
The manuscript is reasonably well written, and the methods seem to me to described with an appropriate degree of detail. The article does require close reading, however, particularly by individuals who are not familiar with the coagulation cascade (though this is briefly introduced in the introduction). There are a few relatively minor issues that should be addressed for clarity and improvement.
1) On page 5, line 203, and in Table 1, the clotting time (CT) of the control group mice is written as 137.17s in the text, but 173.17s in the table. Could the authors clarify which is correct?
2) The dosing of Velefibrinase used in the study was not clearly mentioned. Would the authors clearly define why specific concentrations of 0.22, 0.44, and 0.88 mg/kg were chosen? Additionally, it would be helpful to clarify how these concentrations were classified as low, medium, and high, respectively.
3) The authors could include a short definition of the coagulation tests aPTT, PT, and TT for readers who are not familiar with the field so that they understand better what is being tested.
4) The authors reference their previous study (Zhou et al., Microorganisms, 2022) on Bacillus velezensis Velefibrinase, wherein the enzyme dose-dependently increased the activated partial thromboplastin time (aPTT). This reveals that Velefibrinase influences both the intrinsic and common coagulation pathways of blood, transferring its anticoagulant activity. From this, the authors, on page 12 (lines 398-403), postulate that Velefibrinase prevents Factor XII (Hageman factor) from acting in vivo, offering one explanation for the consequent increase in clotting time.
Although the prolonged aPTT indicates Velefibrinase acts on the intrinsic coagulation pathway, additional clarification is needed. Since Factor XII is the activator for the intrinsic pathway, a prolongation of aPTT would mean Velefibrinase acts upon FXII activation or possibly on factors further downstream within the cascade. Since Velefibrinase is a fibrinolytic agent that lyses fibrin clots, it will indirectly lower the responses to clotting, and thus induce prolongation of aPTT.
However, while these results imply the involvement of FXII, the authors have not directly demonstrated that Velefibrinase acts on FXII or the other factors of the intrinsic pathway. The argument that the enzyme could inhibit the activation of FXII is plausible, but it is so until the actual experiments are performed on FXII. Moreover, the capacity of Velefibrinase to reduce the availability of fibrinogen may also be responsible for the delayed clotting times, thus rendering its exact mode of action more difficult to understand.
I recommend rewriting some of the text. The authors should be more precise in stating that Velefibrinase inhibits FXII, as it has not been proven yet. They may want to rewrite the manuscript so it is clear that this is a hypothesis from the fact that there is an extended aPTT and it should be explored directly.
Also, further experiments need to be conducted to directly evaluate if Velefibrinase inhibits FXII, the most important factor implicated to be affected. Chromogenic assays may be used to measure FXII activation upon addition of Velefibrinase, providing evidence of inhibition. It will be helpful to test other intrinsic pathway factors, such as FXI, PK, IX, and VIII, using coagulation factor activity tests. Measuring fibrinogen after Velefibrinase treatment would confirm whether its reduction is the cause of the slowed clotting. I would recommend that the authors explore the effects of Velefibrinase on the intrinsic and extrinsic pathways in the future.
Author Response
Response to Reviewer 2 Comments
Point 1: On page 5, line 203, and in Table 1, the clotting time (CT) of the control group mice is written as 137.17s in the text, but 173.17s in the table. Could the authors clarify which is correct?
Respond 1: Thanks for your reminder. The CT of control group mice is 173.17 s, and I have revised it.
Point 2: The dosing of Velefibrinase used in the study was not clearly mentioned. Would the authors clearly define why specific concentrations of 0.22, 0.44, and 0.88 mg/kg were chosen? Additionally, it would be helpful to clarify how these concentrations were classified as low, medium, and high, respectively.
Respond 2: The middle-dosage of Velefibrinase is determined based on the urokinase instructions and calculated by using the Meeh-Rubner formula with reference to body surface area. The low-dosage and high-dosage were then set according to a two-fold increment relationship, with the low-dosage being half the middle-dosage and the high-dosage being twice the middle-dosage. These concentrations classified design is consistent with the practices described in other literatures.
Reference:
Zhao L, Lin X, Fu J, et al. A Novel Bi-Functional Fibrinolytic Enzyme with Anticoagulant and Thrombolytic Activities from a Marine-Derived Fungus Aspergillus versicolor ZLH-1. Mar. Drugs 2022, 20, 356. DOI: 10.3390/md20060356.
Irfan M, Jeong D, Kwon H W, et al. Ginsenoside-Rp3 inhibits platelet activation and thrombus formation by regulating MAPK and cyclic nucleotide signaling [J]. Vascular Pharmacology, 2018, 109: 45-55. DOI: 10.1016/j.vph.2018.06.002.
Point 3: The authors could include a short definition of the coagulation tests aPTT, PT, and TT for readers who are not familiar with the field so that they understand better what is being tested.
Respond 3: Thanks for your reminder. I have made the revisions as required. Moreover, the added text content is as follows: The effect of Velefibrinase on the coagulation cascade was evaluated by detecting the coagulation parameters. The activated partial thromboplastin time (aPTT) reflects the intrinsic coagulation pathway, mainly influenced by factors VIII, IX, XI, and XII. The pro-thrombin time (PT) is associated with the extrinsic coagulation pathway, primarily regulated by coagulation factors V, VII, and X. Meanwhile, the thrombin time (TT) measures the ability of fibrinogen to convert into fibrin, indicating the functional status of the terminal stage of coagulation.
Point 4: I recommend rewriting some of the text. The authors should be more precise in stating that Velefibrinase inhibits FXII, as it has not been proven yet. They may want to rewrite the manuscript so it is clear that this is a hypothesis from the fact that there is an extended aPTT and it should be explored directly.
Respond 4: Thank for your reminder. We have rewritten this part of the content, and it is as follows: In addition, the CT also serves as an indicator of coagulation factor effectiveness in the intrinsic pathway. Our previous research showed that Velefibrinase inhibits κ-carrageenan induced thrombus formation in mice tail [15]. Currently, the prevailing view is that κ-carrageenan activates the Hageman factor (coagulation factor XII) in vivo, thereby initiating the intrinsic coagulation pathway and ultimately leading to thrombus formation [26]. Based on this, we speculate that Velefibrinase may prolong the CT by inhibiting Hageman factor or other coagulation factors in the intrinsic pathway. This observation was also consistent with our findings relating to coagulation parameters, specifically that Velefibrinase prolonged aPTT (Table 4), indicating its potential effect on intrinsic pathway factors (XII, VIII, IX, XI). Considering that Velefibrinase can reduce FIB content and directly lyse fibrin clots, both of which contribute to the prolongation of CT. Therefore, whether Velefibrinase directly affects the activity of intrinsic coagulation factors requires further experimental verification.
Point 5: I would recommend that the authors explore the effects of Velefibrinase on the intrinsic and extrinsic pathways in the future.
Respond 5: Thanks for your reminder. We are currently conducting research on a low-cost fermentation process and will investigate the effects of Velefibrinase on the intrinsic and extrinsic coagulation pathways in future research.
Reviewer 3 Report
Comments and Suggestions for Authors
Great work. You have studied effectively and thoroughly the different mechanisms that contribute to the formation of thrombi, in pulmonary embolism, in myocardial infarction and in cerebral ischemic stroke. The parameters chosen represent well the inflammatory processes involved, the involvement of the hemostasis system, in particular on the platelet and coagulation components and the inflammatory processes. It would have been interesting to be able to also perform a global test such as the thrombin generation test. In any case, congratulations: a great job!
Author Response
Thank you so much for your positive comments. Your support is a huge encouragement and we will strive to maintain the quality of our research.
Round 2
Reviewer 1 Report
Comments and Suggestions for Authors
No further concerns